# Interaction of Drug-Sensitive and -Resistant Human Melanoma Cells with HUVEC Cells: A Label-Free Cell-Based Impedance Study

**DOI:** 10.3390/biomedicines11061544

**Published:** 2023-05-26

**Authors:** Giuseppina Bozzuto, Marisa Colone, Laura Toccacieli, Agnese Molinari, Annarica Calcabrini, Annarita Stringaro

**Affiliations:** National Center for Drug Research and Evaluation, Istituto Superiore di Sanità, Viale Regina Elena 299, 00161 Rome, Italy; marisa.colone@iss.it (M.C.); laura.toccacieli@iss.it (L.T.); agnese.molinari@iss.it (A.M.); annarica.calcabrini@iss.it (A.C.); annarita.stringaro@iss.it (A.S.)

**Keywords:** cancer cell extravasation, label-free cell impedance assay, light and electron microscopy, human melanoma cells, multidrug resistance

## Abstract

Cancer cell extravasation is a crucial step in cancer metastasis. However, many of the mechanisms involved in this process are only now being elucidated. Thus, in the present study we analysed the trans-endothelial invasion of melanoma cells by a high throughput label-free cell impedance assay applied to transwell chamber invasion assay. This technique monitors and quantifies in real-time the invasion of endothelial cells by malignant tumour cells, for a long time, avoiding artefacts due to preparation of the end point measurements. Results obtained by impedance analysis were compared with endpoint measurements. In this study, we used human melanoma M14 wild type (WT) cells and their drug resistant counterparts, M14 multidrug resistant (ADR) melanoma cells, selected by prolonged exposure to doxorubicin (DOX). Tumour cells were co-cultured with monolayers of human umbilical vein endothelial cells (HUVEC). Results herein reported demonstrated that: (i) the trans-endothelial migration of resistant melanoma cells was faster than sensitive ones; (ii) the endothelial cells appeared to be strongly affected by the transmigration of melanoma cells which showed the ability to degrade their cytoplasm; (iii) resistant cells preferentially adopted the transcellular invasion vs. the paracellular one; (iv) the endothelial damage mediated by tumour metalloproteinases seemed to be reversible.

## 1. Introduction

Tumour metastasis is a very complex and dynamic cell biological process characterised by a number of complex interactions between tumour cells and host tissue. Metastasising tumour cells must traverse natural barriers, such as connective tissue components and organ epithelia at multiple stages of metastatic process [1]. Among these events leading to metastatic formation, the process of extravasation, the active migration of tumour cells across the endothelial barrier and penetration of the underlying basement membrane, has long been considered a key step in metastasis [2,3,4].

The extravasation process for single tumour cells can be subdivided into the following steps: (1) adhesion of tumour cells to vascular endothelium, (2) transmigration across the endothelial lining and the underlying basement membrane, and (3) invasion of the surrounding tissue [5,6]. Single tumour cells can adhere to the vascular endothelium first and then grow as a small colony/spheroid as described [7]. On the other hand, small tumour cell colonies/spheroids themselves could be detached from the primary tumour [8], be transported by the blood stream, and finally adhere to the vessel wall [9].

Our current understanding of the sequential steps of tumour cell extravasation is mainly based on in vitro models using modified Boyden chamber/transwell assays [4,10,11]. The model consisted of a confluent monolayer of endothelial cells cultured on coverslips coated with Matrigel^TM^ [12].

These conventional methods are semiquantitative because cells need to be labelled with a fluorescent dye or other dyes either before or after the experiment to measure cell phenotypes. Moreover, these existing methods are time-consuming, labour-intensive, and measure the results at only one time point (end point measurements). In addition, these methods are prone to making inaccurate measurements due to inconsistent handling during the experimental procedure [13].

Unlike conventional methods, the real-time cell analysis system measures cell impedance in real-time without requiring pre- or post-staining and mechanical damage of cells. Another important advantage that this technique offers is that it allows you to extend the experiment duration so that biological effects can be determined in a time-dependent manner. Executing the experiment is time-efficient and not labour-intensive. Label-free cell impedance assay is one of the best real-time measurements to measure cell migration and invasion [13,14,15,16,17].

Thus, in the present study, we analysed the trans-endothelial invasion of melanoma cells by a high throughput label-free cell impedance assay applied to a transwell chamber invasion assay. As described above, this technique monitors and quantifies in real-time the invasion of endothelial cells by malignant tumour cells [17]. In particular, we used human melanoma M14 wild type (WT) cells and their drug-resistant counterparts, M14 multidrug-resistant (ADR) cells, selected by prolonged exposure to doxorubicin (DOX) [18]. As previously published by our group [19,20], drug-resistant M14 ADR cells show a more invasive phenotype compared to parental cells, as demonstrated by quantitative transwell chamber invasion assay. This is accomplished by a different migration strategy adopted by resistant cells previously described in tumour cells with high metastatic capacity [21]. The main aim of the present study was to compare the intimate interaction of drug-sensitive and -resistant melanoma cells with endothelial cells, crucial step in the metastasising phase of this poor prognosis neoplasm. To this purpose, tumour cells were co-cultured with both confluent and sub-confluent monolayers of human umbilical vein endothelial cells (HUVEC). HUVEC cells were chosen as the endothelium model since they are the most characterised and still widely used in tumour cell transmigration studies [22,23,24,25,26,27,28,29].

Results obtained by impedance analysis were compared with those obtained by laser scanning confocal microscopy (LSCM), scanning electron microscopy (SEM), and transmission electron microscopy (TEM). Functional study was also performed by gelatine zymography. Ultrastructural and functional studies completed our study and allowed us to elucidate the mechanism of tumour–endothelial cells interaction. In particular, they gave information about the choice of transmigration strategies on the basis of the molecular ‘weapons’ available to tumour cells, thus allowing us to hypothesise two different behaviours adopted by M14 WT and M14 ADR cells as depicted in the proposed model.

Results herein reported demonstrated that: (i) the trans-endothelial migration in resistant melanoma cells was faster than that in sensitive ones; (ii) the endothelial cells appeared to be strongly affected by the transmigration of melanoma cells which showed the ability to degrade their cytoplasm; (iii) resistant cells preferentially adopted the transcellular invasion vs. the paracellular one; (iv) the endothelial damage mediated by tumour metalloproteinases seemed to be reversible.

## 2. Materials and Methods

### 2.1. Cell Cultures

The established human melanoma cell line M14 WT (UCLA SO-M14), kindly gifted by Dr. G Zupi (Regina Elena Institute for Cancer Research, Rome, Italy) [30], and its derivative multidrug-resistant variant (M14 ADR) were grown in RPMI 1640 medium (Euroclone, Pero, Italy) supplemented with 10% FBS (Hyclone, Carmlington, UK), 100 μg/mL streptomycin, 100 U/mL penicillin (Euroclone), 1% L-glutamine (Euroclone), 1% non-essential amino acid (Euroclone) in a humidified atmosphere of 5% CO_2_ in a water-jacketed incubator at 37 °C. M14 ADR cell line was selected culturing M14 cells in the presence of 40 mM DOX (Adriblastina, Pharmacia & Upjohn S.P.A., Milan, Italy) [18].

Human umbilical vein endothelial cells (HUVEC) were purchased from Promocell (Heidelberg, Germany) and cultured in endothelial cell growth medium (ECGM, Promocell) supplemented with 10% FBS and supplement mix (epidermal growth factor, hydrocortisone, basic fibroblast growth factor (bFGF)) at 37 °C and 5% CO_2_. HUVECs from passages 3 to 5 were used for experiments.

DNA fluorochrome staining with Hoechst 33258 bisbenzimide is commonly used for detection of mycoplasma contamination in cell cultures [31].

### 2.2. Transmission Electron Microscopy

For the analysis of ultrathin sections, melanoma cells were co-cultured with a sub-confluent monolayer HUVEC cells for 3 and 5 h. At the end of culturing, samples were fixed with 2.5% glutaraldehyde in 0.1 M cacodylate buffer (pH 7.2) at room temperature for 1 h. After post-fixation with 1% OsO_4_ in cacodylate buffer at room temperature for 2 h, cultures were dehydrated through graded ethanol concentrations and embedded in Epon 812 resin (Electron Microscopy Science, Fort Washington, PA, USA). Ultrathin sections, obtained with a LKB ultramicrotome (LKB, Bromma, Sweden), were stained with uranyl acetate and lead citrate and examined with a Philips 208S transmission electron microscope (FEI Company, Eindhoven, The Netherlands).

### 2.3. Laser Scanning Confocal Microscopy

For laser scanning electron microscopy study, 1.8 × 10^4^ HUVEC cells/cm^2^ were seeded on 12 mm glass coverslips. The HUVECs were left to attach on the coverslip and grow for 18–21 h until they formed a monolayer. Once a HUVEC monolayer formed, melanoma cells were detached, labelled with 1,1′-Dioctadecyl-3,3,3′,3′-tetramethylindocarbocyanine perchlorate (Dil C18 [3]) 12.5 μg/mL in HBSS for 10 min at 37 °C, washed, and added to HUVEC cultures. After 1, 3, or 5 h, co-cultures were fixed in 3.7% paraformaldehyde with 2% sucrose, for 30 min at room temperature. Samples were then washed twice in PBS and permeabilised with 0.5% Triton X-100 for 5 min. Samples were then washed in PBS and incubated with phalloidin FITC (1:100) for 30 min at room temperature. Cells were rinsed three times with PBS afterwards. Finally, all samples were mounted in PBS containing 50% glycerol.

Observations were performed with a Leica TCS SP2 laser scanning confocal microscope (Leica Microsystems, Mannheim, Germany). The excitation wavelengths used were 488 nm and 543 nm for fluorescein and for rhodamine, respectively.

### 2.4. Scanning Electron Microscopy

For scanning electron microscopy (SEM) analysis, cells were grown on 12 mm glass coverslips and treated as above reported. At the indicated times, cells were fixed with 2% glutaraldehyde in 0.1 M cacodylate buffer, pH 7.4 at room temperature for 30 min, postfixed with 1% OsO_4_ in the same buffer, dehydrated through a graded ethanol series, critical point dried with CO_2_, and gold coated by sputtering. Samples were examined with a Cambridge Stereoscan 360 scanning electron microscope (Cambridge Instruments Ltd., Cambridge, UK).

### 2.5. Gelatine Zymography

Gelatine zymography was carried out on diluted supernatants collected from the migration-assay upper and lower compartments as described by [32]. In brief, samples were applied to 7.5% polyacrylamide gels containing gelatine at final concentration of 1 mg/mL. Following electrophoresis, gels was rinsed twice for 30 min in 2.5% Triton X-100, 50 mmol/L Tris/HCl, pH 7.5, and 5 mmol/L CaCl_2_ before being incubated overnight at 37 °C in incubation buffer containing 50 mM Tris-HCl, 0.04 M NaCl, and 5 mM CaCl_2_. Gels were then stained for 3 h in Coomassie Brilliant Blue R-250 followed by destaining with methanol, acetic acid, and water (4:1:5) before being transferred to water. Zones of enzymatic activity were visualised as clear bands against a blue background. Conditioned medium from BHK-21 (Baby hamster kidney cell-21) cells was used as a positive control. The gel was photographed and the image was analysed and quantified using the image analysis program Image J (National Institutes of Health, Bethesda, MD, USA).

### 2.6. Electrical Impedance Assay

Melanoma cell transmigration throughout the HUVEC monolayer was continuously monitored using the impedance-based measuring device xCELLigence RTCA (Real Time Cell Analysis) DP (Dual Purpose) from Agilent (Santa Clara, CA, USA). The xCELLigence station is placed in a 37 °C incubator in the presence of 5% CO_2_ and is constantly connected to a computer (RTCA DP control unit, containing the RTCA software Pro version 2.2) dedicated to measuring the values of impedance. The readings were taken by applying an electric potential of 22 mV at 10 Hz.

For the transmigration experiment, the CIM-Plate 16 was used. The CIM-Plate 16 contains 16 wells that are divided into upper and lower chambers by a microporous polyethylene terephthalate (PET) membrane with 8 µm pores, allowing cells to adhere to microelectrodes on the underside of the membrane [33]. The number of adherent cells causes a change in impedance (Figure 1). The impedance sensors automatically detect cells as they migrate and attach to the impedance microelectrodes in the lower chamber. The impedance of electron flow caused by adherent cells was reported using a unitless parameter called cell index (CI), where CI = (impedance at time point n—impedance in the absence of cells)/impedance value nominal.

In brief, 160 µL HUVEC cell medium was added to lower chamber. The CIM-Plate 16 was assembled by placing the top chamber onto the lower chamber and snapping the two together. Then, 160 µL of medium was placed in the top chamber to hydrate and pre-incubate the membrane for 30 min in the CO_2_ incubator at 37 °C before obtaining a background measurement. Following this step, 50 μL of the HUVEC cell suspension (130.960 cells/mL; confluent monolayer) was carefully added to each well of the upper chamber to avoid bubble formation. The CIM-Plate 16 was left to equilibrate for 30 min and then placed in the RTCA DP-station to begin with the impedance readings. The HUVECs were left to attach to the electrodes and grow for 18–21 h until they formed a monolayer, as evidenced by a flattening of the cell index. Once a stable HUVEC monolayer formed, the CIM-Plate 16 was then removed from the RTCA DP station and 50 μL of M14 WT or M14 ADR cell suspension (261.920 cells/mL) was added to the upper chamber. The CIM-Plate 16 was placed in the RTCA DP station and co-cultures HUVEC/melanoma cells were monitored every 30 min for 120 h.

## 3. Results

### 3.1. Analysis of Melanoma Cell–HUVEC Interaction by Laser Scanning Confocal Microscopy

In order to analyse the interaction of melanoma cells with the human umbilical vein endothelial cells (HUVEC), laser scanning confocal microscopy (LSCM) observations were performed on double-labelled co-cultures. Melanoma cells were labelled with the lipophilic membrane dye 1,1′-dioctadecyl-3,3,3,3′-tetramethylindocarbocyani perchlorate (DiI). Then, melanoma cells tagged with DiI were added to HUVEC cell cultures. The co-cultures were observed after 1, 3, and 5 h of interactions, after fixation and staining of F-actin.

Figure 1 shows serial optical sections (from the top to the bottom) of the drug-sensitive melanoma (M14WT)/HUVEC co-cultures: the red signal comes from tagged melanoma cells, the green signal from fluorescein actin of HUVEC cells, and the yellow signal reveals the interaction of tumour cells with endothelial cells. Within 1 h, several M14 WT invasive protrusions at the initial stage of attachment to endothelial cells (EC) (yellow signal) were visible. A few melanoma cells were beginning to penetrate the EC, as revealed by the red signal. After 3 h of interaction, sensitive melanoma cells appeared to infiltrate both between adjacent HUVEC cells (intercellular invasion) (Figure 2, arrows) and within isolated cells (Figure 2, arrow heads) (transcellular invasion). At the initial stage of attachment to the endothelium, melanoma cell morphology appeared round. They soon underwent drastic changes in cell shape during trans-endothelial transmigration, as confirmed by SEM images (see Section 3.2). Observations carried out on M14WT/HUVEC cell co-culture after 5 h of interaction (Figure 3) displayed melanoma cells which completed the transmigration process: in the image, clustered M14 WT cells (arrow heads) undergoing transcellular invasion are shown. During transcellular invasion, the HUVEC F-actin was assembled into a ring-like array around the circumference of the invasion pore (arrows) that seems to encapsulate the invading cancer cells. Moreover, the actin network of endothelial cells appeared dramatically altered.

In Figure 4 and Figure 5, M14 ADR/HUVEC cell co-cultures are shown. LSCM observations indicated that M14 ADR resistant cells displayed a higher ability to attach to HUVEC cells compared to their parental counterparts. Moreover, resistant cells were able to complete the trans-endothelial migration after 3 h, whereas M14 WT cells took 5 h (Figure 3). Even in this case, a ring-like actin array around the circumference of the invasion pore was clearly detectable (Figure 5, arrow).

Thus, the results obtained by LSCM indicated that both drug-sensitive and -resistant cells displayed the capability of invading the endothelial monolayer; however, resistant cells seemed to adhere to HUVEC cells more efficiently. Moreover, confocal microscopy observations strongly suggest a strict cooperation between HUVEC and melanoma cells during the transmigration process.

### 3.2. Electron Microscopy Observations

Scanning electron microscopy (SEM) analysis, allowing the study of cell interaction at high resolution, revealed that, nevertheless, at the initial stage (1 h) of attachment to the endothelium, both sensitive and resistant melanoma cells interacted with endothelial cells (Figure 6 and Figure 7, respectively). Tumour cells appeared rounded in shape, with the surface covered by randomly distributed microvilli. Moreover, in agreement with previous studies [12], numerous membrane blebs protruding from the basolateral surfaces of sensitive melanoma cells were visible (Figure 6b). Contact regions also showed thin microvilli arising from the underlying endothelial cells (Figure 6c). SEM analysis also allowed us to visualise resistant melanoma cells prolonging cellular protrusions that pierced the cytoplasm of endothelial cells (Figure 7b,c). These protrusions extended laterally from the basal end of the melanoma cells and they were not usually visible, suggesting that their formation may be induced by the interaction between melanoma cells and endothelial cells. Interestingly, a number of small vesicles appeared attached on protrusions (Figure 7c), suggesting that exoplasmic vesicles containing proteinases might be released in proximity of the contact regions. In addition, in M14 ADR/HUVEC co-cultures (Figure 7a), endothelial cells attached by resistant melanoma cells, after 1 h of interaction, displayed ruptures at cell surface and holes in the cytoplasm to a greater extent than HUVEC cells in the co-cultures with M14 WT cells (Figure 6a).

When compared to HUVEC/M14 WT co-cultures (Figure 8), after 3 h of incubation a higher number of melanoma cells were visible in HUVEC/M14 ADR samples (Figure 9). Endothelial cells interacting with resistant melanoma cells displayed a dramatically altered morphology (Figure 9). In addition, both M14 WT and M14 ADR cells were able to degrade the cytoplasm of HUVEC cells, confirming that the trans-endothelial migration occurred both through the inter-endothelial junctions and transcellularly through the cytoplasm of the endothelial cells. The cooperation of endothelial cells in the interaction with tumour cells was further confirmed by the protrusion of HUVEC cell embracing melanoma cell (Figure 8b).

After 5 h of incubation with M14 WT cells, endothelial cells still appeared damaged (Figure 10). In particular, melanoma cells passing through the cytoplasm of a HUVEC cell were visible (Figure 10b). As resistant melanoma cells came into contact with the substrate, they spread on the surface and adopted a fibroblastic morphology, becoming sandwiched (Figure 11). In M14 ADR/HUVEC co-cultures, when the migration of tumour cells was completed, endothelial cells seemed to recover their damaged morphology.

Transmission electron microscopy confirmed the transcellular invasion process (Figure 12). Melanoma cells completely immersed in the cytoplasm of HUVEC cells could be observed (Figure 12a–c). Tumour cells were recognisable for cell diameter and chromatin organisation. The presence of eterochromatin, that appears more electron-dense than euchomatin, is a characteristic of migrating tumour cells [34,35]. This result definitely confirmed LSCM observations. Moreover, adhering tumour cells seemed to be able to degrade the cytoplasm of endothelial cells (Figure 12d–f), accounting for the cell wounds observed by SEM.

Thus, (i) trans-endothelial migration of melanoma cells was accomplished by both intercellular and transcellular invasion; (ii) these processes most likely were mediated by metalloproteinase; (iii) endothelial cells seemed to recover the damage after the passage of drug-resistant melanoma cells.

### 3.3. Gelatine Zymography

Matrix metalloproteinases (MMPs), a family of Zn-endopeptidases, are implicated in cell movement during development, angiogenesis, tissue remodelling, and tumorigenesis [5]. A subgroup of the MMP family, the gelatinases or type IV collagenases, display a particular specificity for BM collagens [6]. The involvement of two gelatinases, A (MMP2) and B (MMP9), particularly gelatinase B, in extravasation of inflammatory cells has been postulated (reviewed in [7]).

Thus, to verify the hypothesis in point (ii), gelatine zymography was carried out on diluted supernatants collected from the upper and lower migration-assay compartments as described by Overall and co-workers [32]. The results obtained demonstrated that following the interaction with HUVEC cells, both drug-sensitive and -resistant melanoma cells were able to release MMP9 gelatinase (Figure 13). However, the amount released by resistant cells was higher than that of sensitive ones, after both 5 and 24 h of interaction.

### 3.4. Electrical Impedance Assay Technique

To verify the point (iii) reported above (Section 3.2), we employed an electrical-impedance based technique that monitors and quantifies in real-time the invasion of endothelial cells by malignant tumour cells [36].

In particular, the xCELLigence RTCA DP system and the CIM-Plate 16 devices were employed. The instrument measures changes in electrical impedance as cells attach and spread in a culture dish covered with a gold microelectrode array that covers approximately 80% of the area on the bottom of a well. The attachment and spreading of cells on the electrode surface leads to an increase in electrical impedance. The impedance is reported as cell index, a dimensionless parameter, which is directly proportional to the total area of tissue-culture well that is covered by cells. Hence, the cell index can be used to monitor cell adhesion, spreading, morphology, and cell density. Cell index curve trends (morphology) give specific information about the cell growth (or death) and are specific for each cell line. The invasion assay described in this study is based on changes in electrical impedance at the electrode/cell interphase, as a population of malignant cells invade through a HUVEC monolayer. The data were obtained in real-time and were more easily quantifiable for a long period of incubation, as opposed to end-point analysis for other methods. The co-cultures HUVEC/melanoma cells were followed up to 120 h after the inoculation of tumour cells on the endothelial monolayer (Figure 14).

Changes in cell index were revealed as HUVEC cells attached to and spread on CIM plates. The formation of the confluent monolayer was represented by a stabilisation of cell index (each value represents the mean of two-well cell indexes) after 18 h from the seeding. This phase was followed by stabilisation at a higher cell index for up to another 20 h. After that, the HUVEC cell index decreased up to 1.2 cell index value, indicating the survival of the culture.

When HUVEC cells were challenged with M14 WT and M14 ADR cells (Figure 14, arrow), there was a drop in electrical resistance and consequently of the cell index (each value represents the mean of three-well cell indexes). This drop represented the invasion of the HUVEC monolayer by the invading tumour cells [13]. Of interest is that the curve morphologies related to M14 WT and M14 ADR/HUVEC co-cultures were coincident up to 70 h. After this time, the cell index values of drug-sensitive M14 WT/HUVEC co-cultures increased again, indicating that some drug-sensitive tumour cells were still in the upper chamber and survived over the time, very likely taking advantage of the debris coming from endothelial cells. Differently, a stabilisation of the cell index values related to M14 ADR cells/HUVEC co-cultures was observed, following and finally overlapping HUVEC culture values. This result indicates that drug-resistant tumour cells more efficiently passed in the lower chamber after the invasion through the endothelial monolayer that seemed to survive even after the invasion of tumour cells.

With this high throughput assay, it was possible to follow in real time the melanoma/endothelial cell co-cultures for a long time, avoiding artefacts due to the preparation of the end-point measurements. The results obtained confirm that both drug-sensitive and -resistant melanoma cells showed the capability of invading the endothelial monolayer, and that resistant cells were able to invade HUVEC cells more efficiently, over time. Moreover, a cooperation between HUVEC and melanoma cells was clearly demonstrated by fluorescence and high-resolution imaging studies. Endothelial cells were able to recover the damage due to the passage of drug-resistant melanoma cells, as shown by the cell index values obtained up to 120 h.

## 4. Discussion

The entry of cancer cells into the bloodstream is a critical point in the metastatic cascade [37] and usually leads to poor prognosis [38,39]. Even if cancer cell extravasation is a crucial step in cancer metastasis, it has not been successfully targeted by current anti-metastasis strategies due to the lack of a complete understanding of the mechanisms that control this process [40].

Literature data suggest that cancer cell extravasation is a complicated multi-step process consisting of adhesion of circulating tumour cells onto the endothelium and the following trans-endothelial migration [41,42]. Forming the inner layer of the vascular system, endothelial cells (ECs) facilitate essential physiological processes of the organism. Vascular ECs allow the vessel wall passage of nutrients and diffusion of oxygen from the blood into adjacent cellular structures. ECs control vascular tone and blood coagulation as well as adhesion and transmigration of circulating cells. The versatility of EC functions corresponds to great cellular diversity. Vascular ECs can form extremely tight barriers, thus restricting the xenobiotic passage (BBB barrier) or immune cell invasion, whereas, in other organ systems, the endothelial layer is discontinuous (e.g., liver sinusoids) or fenestrated (e.g., glomeruli in the kidney), for rapid molecular exchange. ECs not only vary between vascular systems or organs, they also commute along the vascular tree and specialised subpopulations of ECs can be found within the capillaries of a single organ. Three main types of capillary EC have been defined: (i) a discontinuous capillary endothelium with intercellular gaps and an interrupted basement membrane, enabling free exchange of molecules (liver, bone marrow); (ii) a fenestrated endothelium that allows for diffusion of fluids and small molecules (kidney, choroid plexus); (iii) a tight continuous endothelium with a continuous basement membrane. Molecules can pass the continuous endothelium by tightly regulated transcytosis (brain, lung, heart) [43]. Several cell lines have been proposed for in vitro models of the vascular systems: HUVEC, human umbilical vein endothelial cell; iPSC-EC, induced pluripotent stem cell-derived endothelial cell; HDMVEC, human dermal microvascular endothelial cell; HPMEC, human pulmonary microvascular endothelial cell; HBMVEC, human brain microvascular endothelial cell; BM-MSC, HCME3/D3, human cerebral microvascular endothelial cell; and HLESC, human liver sinusoidal endothelial cell. Among these, HUVEC proved to be the model most suitable to study both cardiovascular diseases [44] and extravasation studies of tumour cells such as melanoma [45]. HUVEC cells also displayed the ability to grow on gold electrodes of the CIM plates employed in our study for the electrical impedance assay.

Various biochemical and physical factors play a role in this process in an inter-correlated manner, including multiple cell adhesive molecules (selectins, cadherins, and integrins), chemokines, growth factors, and mechanical factors [46,47]. It has been demonstrated that the vascular permeability can be modulated by cancer cells through either direct contact or secreted growth factors, chemokines, and small extracellular vesicles [47,48,49,50]. Furthermore, endothelial cells have been proven to actively facilitate tumour transmigration. Cancer cells can alter the biomechanical properties of ECs, which may in turn regulate cancer cells’ invasive potential [46,47]. In addition, cancer cells also directly modulate the biochemical signalling within the underlying endothelium [40].

In our experimental conditions, we observed that sensitive and resistant melanoma cells are capable of breaching the endothelial monolayer through two distinct mechanisms: (i) paracellular invasion, wherein cancer cells disrupt the EC border; (ii) transcellular invasion in which the cancer cells penetrate individual ECs without causing cell death [51,52].

The first step of melanoma–HUVEC interactions is the formation of a stable attachment to the endothelial cells, through surface protrusive structures, invadopodia, microvilli, and blebs at the leading edge of invading cells. These structures are the main motor for cellular locomotion and invasion and consist of structural proteins (cortactin, N-WASP, Tks4, Tks5), as well as pericellular proteases such as MMP9 and MMP2 [53]. MMP2 and MMP9 could be released in proximity of the contact regions also by the small exoplasmic vesicles observed on protrusions of melanoma cells. As in other members of the MMP family, MMP2 and MMP9 gelatinases are produced and secreted in latent proenzyme form and activated extracellularly (reviewed in [7]). The involvement of MMP9 in extravasation of inflammatory cells has been proposed. Accordingly, in the present study, the enriched proteases allowed invadopodia of melanoma cells to degrade the HUVEC cytoplasm, favouring the invasion and trans-endothelial migration, principally of drug-resistant M14 ADR cells. Indeed, the higher ability in endothelial invasion displayed by drug-resistant melanoma cells as compared to sensitive ones was related to the higher amount of gelatinases produced and released by M14 ADR cells along with the strong alterations observed in HUVEC cell surface morphology in co-cultures with resistant melanoma cells. These findings were supported by LSCM, SEM, and TEM analyses.

The greater efficiency of drug-resistant cells in crossing the HUVEC cell monolayer could also be explained by an increased activation of intracellular signalling. Results reported in this study are consistent with our previous studies in which we demonstrated that M14 ADR displayed a more invasive phenotype, compared with M14 WT, associated with a higher expression of VLA5 (α5β1) and VLA2 (α2β1) integrins, and CD44 molecule. In particular, the more aggressive phenotype of M14 ADR cells was strictly correlated to the overexpression of the ABC transporter P-glycoprotein that cooperates with CD44 through the activation of ERK ½ and p38 mitogen-activated protein kinase (MAPK) proteins. This activation led to an increase of metalloproteinases (MMP2, MMP3, and MMP9), m-RNAs, and proteolytic activities. In particular, the increased invasive behaviour of M14 ADR cells was associated to the activation of ERM, AKT, ERK, p38, and FAK proteins under migration stimulus [20,21].

Tremblay et al. demonstrated that the activation of p38 and ERK enhances trans-endothelial permeability and migration of HT-29 cells. They also obtained evidence suggesting that p38-mediated increase in trans-endothelial migration of cancer cells depends on a myosin light chain phosphorylation-mediated formation of stress fibres [54]. Similarly, M14 ADR cells can benefit from their molecular pattern in trans-endothelial migration.

Integrins are transmembrane receptors that facilitate cell–cell and cell–extracellular matrix (ECM) interaction that demonstrated a role in the transmigration of cancer cells through a monolayer of endothelial cells [55]. Members of the ezrin–radixin–moesin (ERM) family of proteins are implicated in several aspects of cell motility by acting both as crosslinkers between the membrane, receptors, and the actin cytoskeleton, and as regulators of signalling molecules that are involved in cell adhesion, cell polarity, and migration [56]. P-glycoprotein’s association with the actin cytoskeleton through ERM proteins represents a key factor in conferring to tumour cells a multidrug-resistant phenotype [57]. Moreover, these proteins appear to play a crucial role in the invasion of M14 ADR cells [58].

As mentioned earlier, the transmigration process requires mutual interactions between the tumour and the endothelial cells in which the role of the endothelial monolayer is thought to be pivotal in that it can actively regulate metastasis formation by either allowing or blocking the adhesion, and possibly transmigration, of tumour cells [55,56,59].

Several groups have shown that the endothelial cell apical surface shows relevant membrane dynamics upon initial interaction with transmigrating cells [52,60,61,62]. These events provide evidence of the signalling crosstalk between the transmigrating cancer cell and the endothelial cell that reflect on cytoskeletal and membranous networks. In particular, Khuon et al. have reported that interaction with cancer cells induces localised MLCK activity and contractile function in the underlying endothelial cells [52]. However, several aspects of the endothelial cell response to transcellular invasion and the underlying molecular mechanism remain poorly understood.

Our observations confirm the hypothesis that endothelial cells play an active role in the transmigration process. In fact, as the transcellular invasion proceeds, the HUVEC F-actin begins to reorganise and form ring-like structures that encapsulate the invading cancer cells. Thus, HUVEC cells actively reorganise the cytoskeleton to accommodate the transmigration and circumscribe the invasion.

It has been proposed that the endothelial cell cytoskeleton is an important self-protective mechanism and the underlying biomechanics are a key determinant of trans-endothelial migration. Therefore, the remodelling of the endothelial cell during transcellular diapedesis is a protective response that begins during the active transmigrating phase in the form of tension-driven cytoskeletal restructuring. These events are transient and localised. Indeed, in the late phase of diapedesis upon the exit of the transmigrating cell, the formation of an actin-rich structure seems to seal the pore or “micro wounds” of the endothelial cell as a form of recuperative response [61,63]. This mechanism could explain the recovery of damaged endothelial cells that we observed after the migration of melanoma cells was completed.

The label-free cell-based impedance assay allowed us to follow the melanoma/endothelial cell co-cultures in real time and for a prolonged time, avoiding artefacts due to the preparation of the end-point measurements. The comparison with the functional and ultrastructural data gave information about the choice of strategies on the basis of the molecular ‘weapons’ available to tumour cells. Following the model proposed in the present paper (Figure 14b, scheme), in our experimental conditions, drug-resistant melanoma cells, due to a more aggressive phenotype, migrated successfully and transcellularly through the endothelial monolayer. This allowed the restoration of the endothelial barrier. Wild type melanoma cells most likely preferred the paracellular migration and appeared less efficient in the trans-endothelial migration relative to resistant cells, and in agreement with data discussed before [20,21]. Moreover, due to the permanence on the upper chamber of the label-free transwell chamber assay, they grew, as demonstrated by the increase of the cell index values and the modification of the curve trend.

In conclusion, as also described in our previous studies [64], a high throughput label-free cell impedance-based assay can furnish details of biological phenomena, in real time and for a prolonged time, otherwise not detectable in end-point measurements.

## Data Availability

Not applicable.

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
