# Peer review of "Interaction of Drug-Sensitive and -Resistant Human Melanoma Cells with HUVEC Cells: A Label-Free Cell-Based Impedance Study"

_biomedicines, 2023, doi:10.3390/biomedicines11061544_

Round 1
Reviewer 1 Report
My main problem is about the choice of the study model, i.e. HUVEC that can be more or less sentitive to trans-endothelial invasion of melanoma cells, in particular in comparison to other endothelial cells as well as in the body (physiological conditions) as in vitro, e.g. pulmonary endothelial cells, artery, vein, microvascular endothelial cells. It is well known that the endothelial cells from different origin in vascular tree express various proteins at different levels, and in particular on plasma membrane level.
At least, this point must be discussed with accurate references on endothelium variability
no comments
Author Response
Fist of all, we would like to thank the reviewer for its careful reading of the article and comments.
In the section that follows we detailed our responses, and action taken. Modified text has been underlined in the manuscript.
Generally, following the suggestions, we improved the background in the introduction, with particular regard to the label free cell based technology. Also methods have been described more in detail. We hope that research design and experimental procedure are now more clear. Accordingly, references have also been implemented.
Specific comment:
My main problem is about the choice of the study model, i.e. HUVEC that can be more or less sentitive to trans-endothelial invasion of melanoma cells, in particular in comparison to other endothelial cells as well as in the body (physiological conditions) as in vitro, e.g. pulmonary endothelial cells, artery, vein, microvascular endothelial cells. It is well known that the endothelial cells from different origin in vascular tree express various proteins at different levels, and in particular on plasma membrane level.
At least, this point must be discussed with accurate references on endothelium variability
Response. As reported in the introduction and discussion, HUVEC cells were chosen as endothelium model since is the most characterized and still widely used in tumor cell transmigration studies, according to following references (e.g.).
Mannion, A.J. Live Cell Imaging and Analysis of Cancer-Cell Transmigration Through Endothelial Monolayers. Methods Mol Biol 2022, 2441, 329-338. doi: 10.1007/978-1-0716-2059-5_26.
Casali, B.C.; Gozzer, L.T.; Baptista, M.P.; Altei, W.F.; Selistre-de-Araújo, H.S. The Effects of αvβ3 Integrin Blockage in Breast Tumor and Endothelial Cells under Hypoxia In Vitro. Int J Mol Sci 2022, 23,1745. doi: 10.3390/ijms23031745.
Pietrovito, L.; Leo, A.; Gori, V.; Lulli, M.; Parri, M.; Becherucci, V.; Piccini, L.; Bambi, F.; Taddei, M.L.; Chiarugi, P. Bone marrow‐derived mesenchymal stem cells promote invasiveness and transendothelial migration of osteosarcoma cells via a mesenchymal to amoeboid transition. Mol Oncol 2018, 12, 659–676. doi: 10.1002/1878-0261.12189.
Piwowarczyk, K.; Kwiecień, E.; Sośniak, J.; Zimoląg, E.; Guzik, E.; Sroka, J.; Madeja, Z.; Czyż, J. Fenofibrate Interferes with the Diapedesis of Lung Adenocarcinoma Cells through the Interference with Cx43/EGF-Dependent Intercellular Signaling. Cancers (Basel). 2018, 10, 363. doi: 10.3390/cancers10100363.
Arvanitis, C.; Khuon, S.; Spann, R.; Ridge, K.M.; Chew, T-M. Structure and biomechanics of the endothelial transcellular circumferential invasion array in tumor invasion. PLoS One 2014, 9, e89758. doi: 10.1371/journal.pone.0089758.
Peyri, N.; Berard, M.; Fauvel-Lafeve, F.; Trochon, V.; Arbeille, B.; Lu, H.; Legrand, C.; Crepin, M. Breast tumor cells transendothelial migration induces endothelial cell anoikis through extracellular matrix degradation Anticancer Res 2009, 29, 2347-2355.
Mierke. C.T.; Zitterbart, D.P.; Kollmannsberger, P.; Raupach, C.; Schlötzer-Schrehardt, U.; Goecke, T.W.; Behrens, J.; Fabry, B. Breakdown of the endothelial barrier function in tumor cell transmigration. Biophys J 2008, 94, 2832-2846. doi: 10.1529/biophysj.107.113613.
Leroy-Dudal, J.; Demeilliers, C.; Gallet, O.; Pauthe, E.; Dutoit, S.; Agniel, R.; Gauduchon, P.; Carreiras, F. Transmigration of human ovarian adenocarcinoma cells through endothelial extracellular matrix involves alpha V integrins and the participation of MMP2. Int J Cancer 2005, 114, 531-543. doi: 10.1002/ijc.2077.
Medina-Leyte, D.J.; Domínguez-Pérez, M.; Mercado, I.; Villarreal-Molina, M.T.; Jacobo-Albavera, L. Use of Human Umbilical Vein Endothelial Cells (HUVEC) as a Model to Study Cardiovascular Disease: A Review. Appl Sci 2020, 10, 938. doi: 10.3390/app10030938.
Kim, S.; Wan, Z.; Jeon, J.S.; Kamm, R.D. Microfluidic vascular models of tumor cell extravasation. Front Oncol 2022, 12, 1052192. doi: 10.3389/fonc.2022.1052192.
The endothelium variability has been discussed on the basis of recent literature data (see Discussion section).
Reviewer 2 Report
1. Please complete the basic information about cell lines. Where do they come from (place of purchase), were they authenticated, because the given literature source is from 2004 - are these the same lines?
2. Were the cells analysed for mycoplasma? If so, please indicate the test used.
3. Please describe the methodology in detail so that the whole reaction process is transparent. For example, there is no data on actin staining antibodies - I take it these are actin filaments?
4. „Then, melanoma cells tagged with DiI were added to HUVEC cell cultures. The cocultures were observed after 1, 3 and 5 hours of interactions, after fixation and staining of F-actin” There is a lot of ambiguity in this information for me. First of all - in the materials and methods there is only information about 3 and 5 hours of co-culture - hence the analysis after 1 hour? When HUVEC was stained for F-actin? The description shows that after this 1 h, 3h, and 5h, i.e. when the cancer cells were in co-culture, why did F-actin not stain in the melanoma cells?
5. In my opinion, if the monolayer of the HUVEC line is to reflect the continuity of the endothelium, it should be between 90-100% confluence, and in Fig. 2 there are a lot of empty spaces.
6. On what basis were the cells differentiated in transmission electron microscope images? I suggest using markers.
7. The research methodology and subsequent descriptions of the results obtained are unclear and inconsistent to me.
In connection with the above, I recommend the article to be accepted for publication after major revisions.
Please correct minor grammatical errors
Author Response
First of all, we would like to thank the Reviewer for its careful reading, and criticisms that allowed us to improve the paper. We hope that it is now acceptable for the publication.
In the section that follows we detailed our responses, and action taken. Modified text has been underlined in the manuscript.
- Please complete the basic information about cell lines. Where do they come from (place of purchase), were they authenticated, because the given literature source is from 2004 - are these the same lines?
Response 1. We completed the information about cell line in 2.1 Cell cultures section:
The established human melanoma cell line M14 WT (UCLA SO-M14) , kindly gifted by Dr. G Zupi, (Regina Elena Institute for Cancer Research, Rome, Italy.) Greco C, Zupi G. Biological features and in vitro chemosensitivity of a new model of human melanoma. Anticancer Res. 1987 Jul-Aug;7(4B):839-44. PMID: 3674770. and its derivative multidrug-resistant variant (M14 ADR) were grown in RPMI 1640 medium (Euroclone) supplemented with 10% FBS (Hyclone, Carmlington, UK), 100 μg/ml streptomycin, 100 U/ml penicillin (Euroclone), 1% L-glutamine (Euroclone), 1% non essential aminoacid (Euroclone) in a humidified atmosphere of 5% CO2 in a water-jacketed incubator at 37 °C. M14 ADR cell line was selected culturing M14 cells in the presence of 40 mM DOX (Adriblastina, Pharmacia & Upjohn S.P.A., Milan, Italy) [17]. Human umbilical vein endotelial cells (HUVEC) (Promocell) were grown in PromoCell Growth Medium and subcultivated following the instruction manual from (Promocell).
Human umbilical vein endotelial cells (HUVEC) were purchased from Promocell (Heidelberg, Germany) and cultured in endothelial cell growth medium (ECGM, Promocell) supplemented with 10% FBS and supplement mix (epidermal growth factor, hydrocortisone, basic fibroblast growth factor (bFGF)) at 37 °C and 5% CO2 . HUVECs from passages 3 to 5 were used for experiments.
DNA fluorochrome staining with Hoechst 33258 bisbenzimide is commonly used for detection of mycoplasma contamination in cell cultures.
- Were the cells analysed for mycoplasma? If so, please indicate the test used.
Response 2. DNA fluorochrome staining with Hoechst 33258 bisbenzimide was used for detection of mycoplasma contamination in cell cultures.
Battaglia MF, Balducci L, Finocchiaro M Use of Hoechst 33258 fluorochrome for detection of mycoplasma contamination in cell cultures: development of a technique based on simultaneous fixation and staining..Boll Ist Sieroter Milan. 1980 May 31;59(2):155-8.
Volokhov, D.V.; Graham, L.J.; Brorson, K.A.; Chizhikov, V.E. Mycoplasma testing of cell substrates and biologics: Review of alternative non-microbiological techniques. Mol. Cell Probes 2011, 25, 69–77.
Drexler, H.G.; Uphoff, C.C. Mycoplasma contamination of cell cultures: Incidence, sources, effects, detection, elimination, prevention. Cytotechnology 2002, 39, 75–90.
Young, L.; Sung, J.; Stacey, G.; Masters, J.R. Detection of Mycoplasma in cell cultures. Nat. Protoc. 2010, 5, 929–934.
- Please describe the methodology in detail so that the whole reaction process is transparent. For example, there is no data on actin staining antibodies - I take it these are actin filaments?
Response 3. Required information have been added in the modified in 2.3 laser scanning confocal microscopy section. In particular:
2.3 Laser scanning confocal microscopy
For laser scanning electron microscopy study, 1,8*104 HUVEC cells/cm2 were seeded on 12 mm glass coverslips. The HUVECs were left to attach on the coverslip and grow for 18-21 hours until they form a monolayer. Once a HUVEC monolayer has formed, melanoma cells were detached, labelled with 1,1′-Dioctadecyl-3,3,3′,3′-tetramethylindocarbocyanine perchlorate (Dil C18 (3)) 12.5 μg/ml in HBSS for 10 minuti a 37°C, washed and added to HUVEC cultures. After 1, 3 or 5 hours co-coltures were fixed in 3.7% paraformaldehyde with 2% sucrose for 30 minutes at room temperature. Samples were then washed twice in PBS and permeabilized with 0.5% Triton X-100 for 5 min. Samples were then washed in PBS and incubated with phalloidin FITC (1:100) for 30 minutes at room temperature. Cells were rinsed three times with PBS afterwards. Finally, all samples were mounted in PBS containing 50% glycerol.
Observations were performed with a Leica TCS SP2 laser scanning confocal microscope (Leica Microsystems, Mannheim, Germany). The excitation and emissions wavelengths used were 488 nm and 543 nm for fluorescein and for rhodamine, respectively.
- Then, melanoma cells tagged with DiI were added to HUVEC cell cultures. The cocultures were observed after 1, 3 and 5 hours of interactions, after fixation and staining of F-actin” There is a lot of ambiguity in this information for me. First of all - in the materials and methods there is only information about 3 and 5 hours of co-culture - hence the analysis after 1 hour? When HUVEC was stained for F-actin? The description shows that after this 1 h, 3h, and 5h, i.e. when the cancer cells were in co-culture, why did F-actin not stain in the melanoma cells?
Response 4. Required information have been added in the modified 2.3 laser scanning confocal microscopy section. In transmigrating melanoma cells F-actin (labelled by FITC phallodin) is no detectable in merged images, due to the strong signal of DiI.
- In my opinion, if the monolayer of the HUVEC line is to reflect the continuity of the endothelium, it should be between 90-100% confluence, and in Fig. 2 there are a lot of empty spaces.
Response 5. The image in figure 2 shows a co-culture in which the HUVEC cells were at about 90-100% confluency. Although the field comes from an area of the slide where there are empty spaces between the HUVEC cells, we have chosen to show it because it is representative of the different mechanisms of interaction (intercellular and transcellular invasion) of melanoma cells with endothelial cells.
- On what basis were the cells differentiated in transmission electron microscope images? I suggest using markers.
Response 6. The correlation of the images obtained in transmission electron microscopy with those obtained in confocal laser scanning and scanning electron microscopy allowed us to distinguish between HUVECs and melanoma cells. Moreover, ultrastructural features such as cell size and nuclear organization (i.e. euchromatin and eterochromatin) allowed us to distinguish tumor cells from endothelial cells. We added such information in the manuscript.
- The research methodology and subsequent descriptions of the results obtained are unclear and inconsistent to me.
Response 7. We added information in the introduction, with particular regard to the research methodology. Also methods have been described more in detail in the Material and methods section. We hope that research design, experimental procedure and the obtained results are now more clear. Accordingly, references have also been implemented.
Round 2
Reviewer 2 Report
Most of my comments on the manuscript have been corrected by the authors. In my opinion, the article can be published in this form
Author Response
Thank you for your comments.